# MicroRNAs Influence the Migratory Ability of Human Umbilical Vein Endothelial Cells

**DOI:** 10.3390/genes13040640

**Published:** 2022-04-02

**Authors:** Zhaohui Wang, Ziwei Zeng, Vytaute Starkuviene, Holger Erfle, Kejia Kan, Jian Zhang, Manuel Gunkel, Carsten Sticht, Nuh Rahbari, Michael Keese

**Affiliations:** 1Vascular Surgery, University Clinic Mannheim, Medical Faculty Mannheim, Heidelberg University, 68167 Mannheim, Germany; zhaohuiwang1003@126.com (Z.W.); ziwei.zeng@bioquant.uni-heidelberg.de (Z.Z.); kejia.kan@medma.uni-heidelberg.de (K.K.); zjshxh2011@163.com (J.Z.); nuh.rahbari@umm.de (N.R.); 2BioQuant, Heidelberg University, 69120 Heidelberg, Germany; holger.erfle@bioquant.uni-heidelberg.de (H.E.); manuel.gunkel@bioquant.uni-heidelberg.de (M.G.); 3Institute of Biosciences, Vilnius University Life Sciences Center, 10257 Vilnius, Lithuania; 4NGS Core Facility, Medical Faculty Mannheim, Heidelberg University, 68167 Mannheim, Germany; carsten.sticht@medma.uni-heidelberg.de; 5European Center for Angioscience, Medical Faculty Mannheim, Heidelberg University, 68167 Mannheim, Germany

**Keywords:** HUVEC, angiogenesis, migratory ability, migration, microRNA

## Abstract

To identify miRNAs that are involved in cell migration in human umbilical vein endothelial cells (HUVECs), we employed RNA sequencing under high glucose incubation and text mining within the databases miRWalk and TargetScanHuman using 83 genes that regulate HUVECs migration. From both databases, 307 predicted miRNAs were retrieved. Differentially expressed miRNAs were determined by exposing HUVECs to high glucose stimulation, which significantly inhibited the migratory ability of HUVECs as compared to cells cultured in normal glucose. A total of 35 miRNAs were found as differently expressed miRNAs in miRNA sequencing, and 4 miRNAs, namely miR-21-3p, miR-107, miR-143-3p, and miR-106b-5p, were identified as overlapping hits. These were subjected to hub gene analysis and pathway analysis using the Kyoto Encyclopedia of Genes and Genomes (KEGG), identifing 71 pathways which were influenced by all four miRNAs. The influence of all four miRNAs on HUVEC migration was phenomorphologically confirmed. miR21 and miR107 promoted migration in HUVECs while miR106b and miR143 inhibited migration. Pathway analysis also revealed eight shared pathways between the four miRNAs. Protein–protein interaction (PPI) network analysis was then performed to predict the functionality of interacting genes or proteins. This revealed six hub genes which could firstly be predicted to be related to HUVEC migration.

## 1. Introduction

Together with cell proliferation and tube formation, cell migration is one of the hallmarks of angiogenesis [1,2]. ECs originate from the successive differentiation of mesodermal cells into hemangioblasts, which leads to the formation of the first vascular structures [3]. For this, the cells have to migrate. EC migration is regulated by haptotatic, chemotactic and mechanotactic factors [4]. It requires the activation of several signaling pathways that converge on cytoskeletal remodeling [5]. Then, it follows a series of events in which the EC extend, contract, and throw their rear toward the front and progress forward [5]. Uncovering the molecular regulators that regulate EC migration is of importance, especially to provide a deeper understanding in diseases related to an abnormal angiogenesis [6,7]. While there is abundent information on the role of genes and their proteins on ECs migration, the role of MicroRNAs (miRNAs) for cell migration is still more elusive [8,9]. 

MiRNAs are small non-coding RNA molecules that contain up to 22 nucleotides which regulate cell homeostasis and cellular function by targeting the 3′ untranslated regions of mRNA transcripts [10]. To model HUVEC migration, cells have been exposed to different concentrations of glucose: Patella et al. [8] described the miRNA signature of human umbilical vein endothelial cells (HUVECs) stimulated with normal-glucose (NG) medium and high-glucose (HG) medium, respectively. From the differentially expressed miRNAs, miR-492 was selected for further analysis and the results showed that miR-492 overexpression repressed proliferation, migration, and tube formation of HUVECs. Moreover, the downregulation of endothelial nitric oxide synthase (eNOS) was accompanied by the abovementioned effects. Furthermore, they found repressed VEGF expression in HUVEC, which suppressed in vivo angiogenesis in a tumor xenograft model. Other groups [9] demonstrated that miR-146a knockdown by transfection of antimiR-146a represses lipopolysaccharide (LPS)-induced cell migration and tube formation in HUVEC by targeting caspase recruitment domain-containing protein 10 (CARD10). Bioinformatic analysis of miRanda algorithms and microarray analysis of immunoprecipitated Ago2 ribonucleoprotein complexes were performed in combination to identify the target genes. 

An increasing number of studies have indicated that a high-glucose environment can suppress angiogenesis by inhibiting EC migration among others through the PI3K/AKT-eNOS pathway and the Sirt1/FOXO3 pathway [11,12,13,14]. Therefore, in this study, we conducted a comparative analysis of miRNA sequencing of HUVECs either exposed to NG or HG medium and miRNAs predicted by test mining that could allow the identification of miRNAs, which directly or indirectly control genes involved in glucose-modulated migration.

## 2. Materials and Methods

### 2.1. Cell Culture

HUVECs were purchased from American Type Culture Collection (Manassas, VA, USA) and maintained in Endothelial Cell Growth Media (ECGM) (Provitro, Berlin, Germany), supplemented with 10% fetal bovine serum (FBS; Thermo Fisher, Karlsruhe, Germany) plus 1% antibiotics (streptomycin and penicillin; Life Technologies, Bend, OR, USA). Cells were cultured at 37 °C in an atmosphere containing 5% CO_2_. HUVECs from passage 3 to 7 were used for further experiments. All experiments were performed with cells in their exponential growth phase.

### 2.2. MiRNA Prediction

To investigate the genes that regulate HUVECs migration, relevant studies were identified via PubMed using the search terms “genes” and “HUVECs migration”. The search had no language restrictions. After confirming the genes that regulate HUVECs migration, the genes were then used for the prediction of targeting miRNAs using the miRWalk database (http://mirwalk.umm.uni-heidelberg.de/ (accessed on 5 December 2019)) and TargetScanHuman (http://www.targetscan.org/vert_72/ (accessed on 5 December 2019)). 

### 2.3. Wound Healing Assay

Following the incubation of HUVECs in normal glucose-ECGM (NG-ECGM; 5.5 mM) and high glucose-ECGM (HG-ECGM; 33.3 mM) for 72 h, a 96-pin scratch tool was used to apply a consistent mechanical scratch wound (0.4 mm width) to the cellular monolayer in each well. The cells were then washed twice with Phosphate buffered saline (PBS) and maintained in NG-ECGM and HG-ECGM under low serum condition (1% FBS). Wound closure was monitored using an automated microscope (Olympus IX81, Munich, Germany) equipped with transmitted light at 0 h and 12 h after scratching cells. Wound closure was determined by the decrease in wound area. ImageJ software (NIH, Bethesda, Rockville, MD, USA) was used to analyze the images.

### 2.4. MiRNA Sequencing

Two flasks of HUVECs from same passage were cultured with NG-ECGM and HG-ECGM for 72 h, respectively. Subsequently, HUVECs were harvested and transferred into RNAlater^TM^ RNA Stabilization Reagent (Qiagen, Hilden, Germany) for homogenization. MiRNAs were isolated using the Allprep RNA isolation kit (Qiagen) according to the manufacturer’s protocol. The RNA integrity number (RIN) was determined using the Agilent Bioanalyzer 2100 Expert (B.02.08.SI648, Agilent, Santa Clara, CA, USA). The RNA samples of HG and NG with RINs ranging from 7 to 10 were sent to the Beijing Genomics Institute (BGI, Shenzhen, China) to perform the miRNA sequencing. Filtered miRNAs were quantified by realigning reads to predicted miRNAs in QuickMIRSeq [15]. To decrease false positives, the data were extensively filtered by joint mapping to the transcriptome and ribosomal RNA using QuickMIRSeq. Sequences were aligned to the reference genome GRCh38.p13. The count data were transformed to log2-counts per million (logCPM) using the voom function from the limma package [16] in R. Differential expression analysis was performed utilizing the limma package. A false positive rate of α = 0.05 with a false discovery rate (FDR) correction was considered as the level of significance. Volcano plots and heatmaps were created using ggplot2 package (version 2.2.1) and the iDEP [17](version 0.91).

### 2.5. Protein–Protein Interaction (PPI) Network 

The hub target genes of each miRNA were selected from the STRING database (https://string-db.org/ (accessed on 15 December 2021)). The Cytoscape software (version 3.8.2) was used to analysis the PPI networks basing on the STRING results. The PPI subnetworks were analyzed by using the plug-in Molecular Complex Detection (MCODE). We defined sub-networks with a level of MCODE scores >5 and number of nodes >20. The higher the degree of connectivity of nodes, the greater the role of network stability; the degree of connectivity of each node was calculated by using the plug-in CytoHubba [18]. The top 5 genes with the highest connectivity were identified as key genes.

### 2.6. KEGG Pathway Analysis

The targeted gene of each miRNA were obtained from the miRWalk database, The hub targeted gene and predicted targeted genes of three and four miRNAs were further applied for functional and enrichment analysis. The relative pathways of each miRNA used the targeted gene to analysis. KEGG (Kyoto Encyclopedia of Genes and Genomes) pathways analysis of the hubgene and targeted gene were analyzed by R software (version 4.1.0) and pathways with *p* < 0.05 (adjusted *p* value) were considered statistically significant [19].

### 2.7. Transfection of siRNAs and miRNAs

Plates were prepared for reverse transfection. siRNA/miRNA transfection solution was added to 96-well plates (Nunc) using a Microlab STAR pipetting robot (Hamilton, Reno, NV, USA) as previously described [20]. Then 3 μL OptiMEM (Invitrogen) containing 0.4 M sucrose was transferred to each well of a 96-well plate. Subsequently, 3.5 μL Lipofectamine 2000 (Invitrogen) was added into each well. 5 μL of the respective siRNA/miRNA stock solution (3 μM or 15 μM) and 7.25 μL of a 0.2% (*w*/*v*) gelatin solution containing 1 × 10^−2^% (*v*/*v*) fibronectin (Sigma-Aldrich, Taufkirchen, Germany) were added and mixed thoroughly. The mixed transfection solution was diluted with H_2_O (1:25); 25 μL of the diluted transfection solution was distributed to each well. HUVECs were then collected, and 12,000 cells, dissolved in 100 μL ECGM, were added into the prepared plates for a 48 h incubation under 37 °C. The cells were then washed twice with DPBS (Invitrogen) and maintained in NG-ECGM (Provitro) containing 1% FBS (Thermo Fisher). Wound closure was monitored by fluorescence microscopy (Olympus Biosystems, Munich, Germany) equipped with transmitted light under 37 °C and 5% CO_2_ at 0 h and 18 h. The images were analyzed by ImageJ software.

### 2.8. Statistical Analysis

All experiments were repeated independently at least three times. Data are presented as means ± standard deviations. Statistical comparisons were analyzed with Student’s *t*-tests or one-way analysis of variance using SPSS statistical software (IBM, version 19.0, Armonk, NY, USA). *p* < 0.05 was considered as statistically significant.

## 3. Results

HUVECs were cultured in NG-ECGM and HG-ECGM, respectively. Here, as expected, exposure to high glucose levels led to an increase in migration as determined by a more rapid closure of the gap (Appendix A). Within 24 h, hardly any cell proliferation was observed, as cell doubling time was determined more than 48 h. (Appendix A). 

To predict which miRNAs regulate HUVECs migration (Figure 1), we used “HUVEC migration” and “genes” as keywords to identify the genes which regulate HUVEC migration using the Pubmed database. We hereby extracted 83 genes (Appendix A), all of which had been reported to influence HUVEC migration, to predict miRNAs that may regulate the respective genes. Therefore, the databases miRWalk and TargetScanHuman were used. A total of 2546 miRNAs were obtained from the miRWalk database and a total of 328 miRNAs were acquired from the TargetScanHuman database. Of these, 307 predicted miRNAs overlapped between the two databases (Figure 2 and Appendix A).

To explore which miRNA expression was up- and down-regulated in respect to the glucose-enhanced migration ability, the miRNA expression was determined by miRNA sequencing. HUVECs were collected after exposure to NG-ECGM and HG-ECGM over 72 h. Thereafter, RNA was extracted and sequenced. The volcano plot demonstrated 31 miRNAs which were differentially expressed (fold change > 2, *p* value < 0.05); out of these, 13 were down-regulated and 22 were up-regulated (Figure 3). All differentially expressed miRNAs are listed in Appendix A. We then compared the results derived from text mining and RNA sequencing. Of the 307 genes derived from text mining analysis, 4 overlapped with the 35 hits derived from RNA sequencing, namely miR-21-3p, miR-107, miR-143-3p, and miR-106b-5p (Figure 4). 

We then tested the transfection efficiency of miRNA mimics by the reverse-solid-phase transfection method. Fluorescently labelled pre-miRNAs and anti-miRNAs efficiently entered HUVECs cells grown in NG-ECGM with nearly each cell being transfected after 24 h of incubation (Appendix A). For validation, HUVECs were transfected with miR-21-3p, miR-107, miR-143-3p, and miR-106b-5p. For control, we first used two miRNAs (miR-200b, and miR-214) which were well known to affect the migration of HUVECs [21,22]. As shown in Appendix A, the wound closure significantly quickened in the group transfected by miR-200 and miR-214 as compared to the controls. Using miRNAs at 3 μM yielded a higher transfection efficiency than at 15 μM. Therefore, 3 μM was used for further experiments. Taken together, we hereby also show that wound healing assays are feasible to confirm the role of miRNAs for the migration of HUVECs

When the four overlapping miRNAs were transfected into HUVECs, miR21 and miR107 led to a significant reduction in wound closure by 20% or more as compared to the negative controls. MiR106b and miR143 had a negative influence on migration. For all 4 miRNAs, a significant effect on HUVEC migration could be confirmed (Figure 5)

To investigate potential pathways that may be regulated by genes related to the four miRNAs, MiRWalk database (version 3.0) was used to identify the target genes of these four miRNAs followed by pathway and process enrichment analysis. Subsequently, the gene lists were underwent KEGG analysis by R software [23]. The top 5 enrichment pathways, based on enrich factor, were then obtained (Appendix A). Subsequently, Venny version 2.1 (https://bioinfogp.cnb.csic.es/tools/venny/ (accessed on 8 January 2022)) was used to determine the overlap between the enrichment pathways of the four miRNAs. The most common four shared pathways were: “Neurotrophin signaling pathway”, “Pancreatic cancer”, “Glioma”, “EGFR tyrosine kinase inhibitor resistance”. (Appendix A.)

In addition, protein–protein interaction (PPI) network analysis was used to predict the functionality of interacting genes or proteins. The target genes of these four miRNAs were predicted by miRWalk, followed by uploading the gene list to String (https://string-db.org/ (accessed on 8 January 2022)). The PPIs with combined scores greater than 0.4 were selected for constructing the PPI networks. Tab-separated values of the PPI network were then downloaded and imported to Cytoscape [24]. Subsequently, the MCC method was chosen by the CytoHubba plugin to identify the hub genes. TP53, STAT3, BCL2L11, CCND1, and CYCS were the top five hub genes of miR-21 target genes (Figure 6). SKP1, ZBTB16, VHL, UBE2E3 and LMO7 were the top five hub genes of miR-143 target genes (Figure 6). UBOX5, WSB1, UBE2B, FBXW11, and UBE2D1 were the top five hub genes of miR-106b target genes (Figure 6). BTRC, ZBTB16, FBXL10, FBXL18 and FBXL20 were the top three hub genes of miR-107 target genes (Figure 6). All the above-mentioned hub genes were combined in a gene list which was then subjected to KEGG analysis by R software [19,25]; hereby, the hub genes and the predicted targeted genes that shared between three or four miRNAs were used to predict related pathways (Figure 7 and Appendix A).

Since differential expression may not be the prerequisite for miRNAs to influence HUVEC migration, we also focused on miRNAs which were predicted by text mining. Here, we five 5 miRNAs which had the highest number of predicted target genes related to HUVEC migration. By transfection of the HUVECs with these miRNAs we showed that miR-93, miR-132, and miR-200b could enhance the HUVECs migration, while miR-133a and miR-195 inhibited HUVECs migration (Figure 8). 

## 4. Discussion

While the role of several miRNAs in the migration of HUVECs has already been characterized using cell biological approaches, few approaches have so far combined text mining and sequencing data. 

In this study, we carried out miRNA sequencing on HUVECs treated with NG-ECGM (HUVEC medium containing 5.55 mM glucose) and HG-ECGM (HUVEC medium containing 33.3 mM glucose which is six times as much glucose as the normal medium) for 72 h. Amongst these 35 differently expressed miRNAs (fold change > 2), 6 miRNAs (hsa-miR-6805-5p, -6511a-5p, 6769a-5p, -6880-5p, -6803-5p, and -6756-3p) were firstly identified. The differential expression of the 14 miRNAs demonstrated by Patella et al. [8] could not be reproduced by our work. The probable reason may as follow: firstly, they used lower concentration of glucose (30 mM), while we utilized 33.3 mM; secondly, high-glucose medium was used to stimulate for a shorter time interval in their experiment (72 h vs. 96 h). 

Four miRNAs (miR-21, miR-107, miR-143, and miR-106b) were identified as overlapping hits from the text-mining screen and miRNA sequencing. All four phenotypes could be confirmed after transfecting HUVECs with the miRNAs. We also showed a wider applicability of the phenotype assay. We chose miRNAs which had been predicted by the highest number of migration related genes by text mining, but which were found not be differentially expressed. For all of them the predicted phenotype could be shown using the phenomorphological assay. This underlines the fact that a miRNA does not have to be differentially expressed to cause biological effects in cells. The miR-107 and miR-106b were also used by Rao et al. and Okamoto et al. to screen the function in prostate cancer cell migration and colon cancer cell migration, their results showed the two miRNAs are not hit in their study [26,27]. Our results presented that miR-107 and miR-106b can influence the HUVECs migration.

Various publications have reported data on the four miRNAs which overlapped, which were consistent with our findings [28,29,30,31]. An et al. [29] found that exosomes overexpressing miR-21 are secreted by adipose-derived stem cells (ADSCs). This promotes tube formation in HUVECs in vitro. In their study, miR-21 overexpression upregulated the expression of hypoxia-inducible factor 1-α (HIF-1α), VEGF, stromal cell-derived factor 1 (SDF-1), p-Akt, p-ERK1/2, along with downregulating the expression of phosphatase and tensin homolog (PTEN), revealing that miR-21-enriched exosomes promoted angiogenesis via phosphorylation of Akt and ERK and upregulation of HIF-1α and SDF-1. Li et al. [30] demonstrated how overexpressing of miR-107 promoted tubular formation and migration of HUVECs via targeting of VEGF165 and VEGF164. This was demonstrated by using the Matrigel assay and a trans-well invasion assay, respectively. Climent et al. [28] found that miR-143 and miR-145 regulate angiogenesis by decreasing the proliferation index of HUVECs. Both miRNAs suppressed their capacity to form vessel-like structures when cultured on Matrigel. As two significant genes for the angiogenic potential of HUVECs, hexokinase II (HKII) and integrin β 8 (ITGβ8) were identified as targets of miR-143 and miR-145 respectively. Maimaiti et al. [31] showed that upregulation of miR-106b leads to fewer tubes while downregulation leads to the opposite, as evidenced by tube formation assays. MiR-106b achieved its anti-angiogenic effect in HUVECs via signal transducer and activator of transcription 3 (STAT3)-involved signaling pathway.

To explore potential pathways that might be regulated by genes related to the four miRNAs, the gene lists were predicted by miRwalk and analyzed by KOBAS [23]. The top five pathways based on the amounts of miRNA-related genes were then obtained. The four-most-common shared pathways were: “Neurotrophin signaling pathway”, “Pancreatic cancer”, “Glioma”, “EGFR tyrosine kinase inhibitor resistance”.

In search of protein–protein interactions (PPIs), we performed a further bioinformatical analysis of our data to find hub genes (genes with high correlation in candidate modules). Three hub genes (BTRC, ZBTB16, and FBXL18) were firstly predicted to be related to HUVEC migration. All the three hub genes are related to protein ubiquitination [32,33,34,35,36,37]. As increasing evidence points towards a link between protein ubiquitination and the migration of HUVECs [38,39], these hub genes represent particularly interesting candidates for further investigation. Restoration of ZBTB16 expression led to inhibition of migration in breast cell line [40]; besides, ZBTB16 were demonstrated to be related to locomotion in human airway epithelial cells [41]. FBXL18 played a role in the migration of bladder cancer cell line [42].

Taken together, this project has two major findings. Firstly, we combined data derived from RNA sequencing and text mining to isolate miRNAs involved in HUVEC migration; four miRNAs were identified. Secondly, we found that miRNA do not have to be differentially expressed to cause a biological effect in cells.

The current study has two limitations. Firstly, cells were followed over 12 h which is significantly lower than the doubling time observed for HUVECs. Since the cells were not synchronized for the migration assays, to some extend cells may also undergo mitosis within the observation period. Thus, gap closure cannot be attributed to migration only, but also to some extend to proliferation. Secondly, since no experimental confirmation of the in silico prediction of the hub genes has been obtained, ongoing studies will have to clearly distinguish between migration and proliferation and focus on the functional validation of the hub genes predicted on the confirmation of the binding sites between the hub genes and corresponding miRNAs.

## Figures and Tables

**Figure 1 genes-13-00640-f001:**
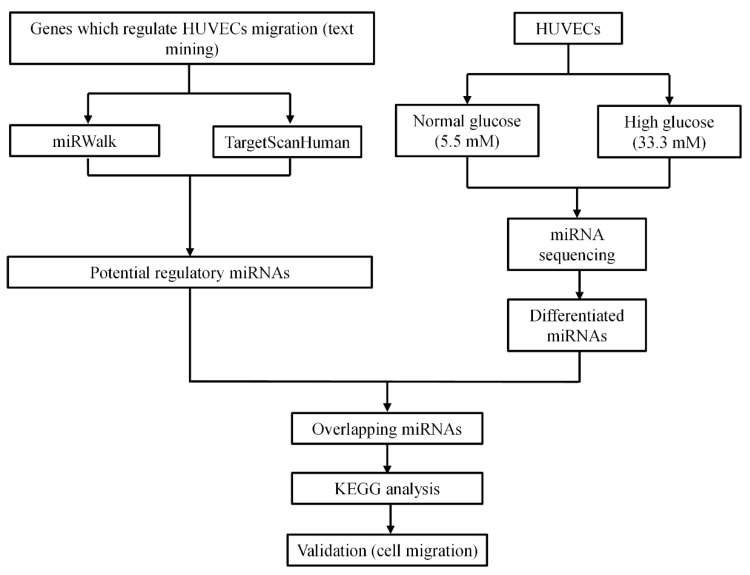
The study scheme.

**Figure 2 genes-13-00640-f002:**
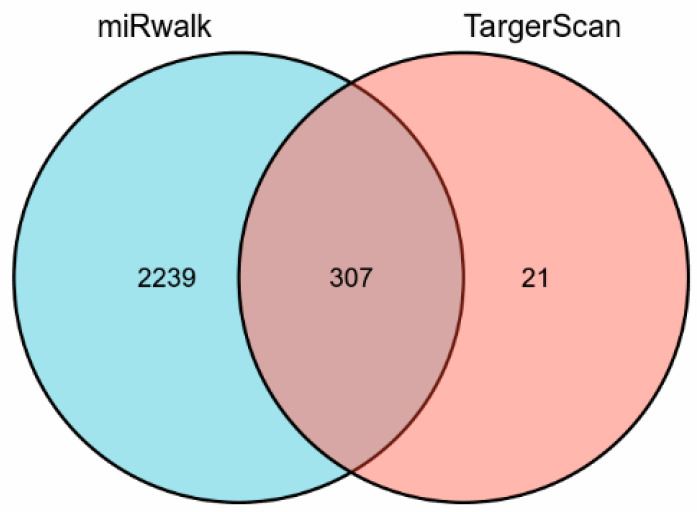
Overlapping predicted miRNAs derived from the miRWalk and TargetScanHuman database.

**Figure 3 genes-13-00640-f003:**
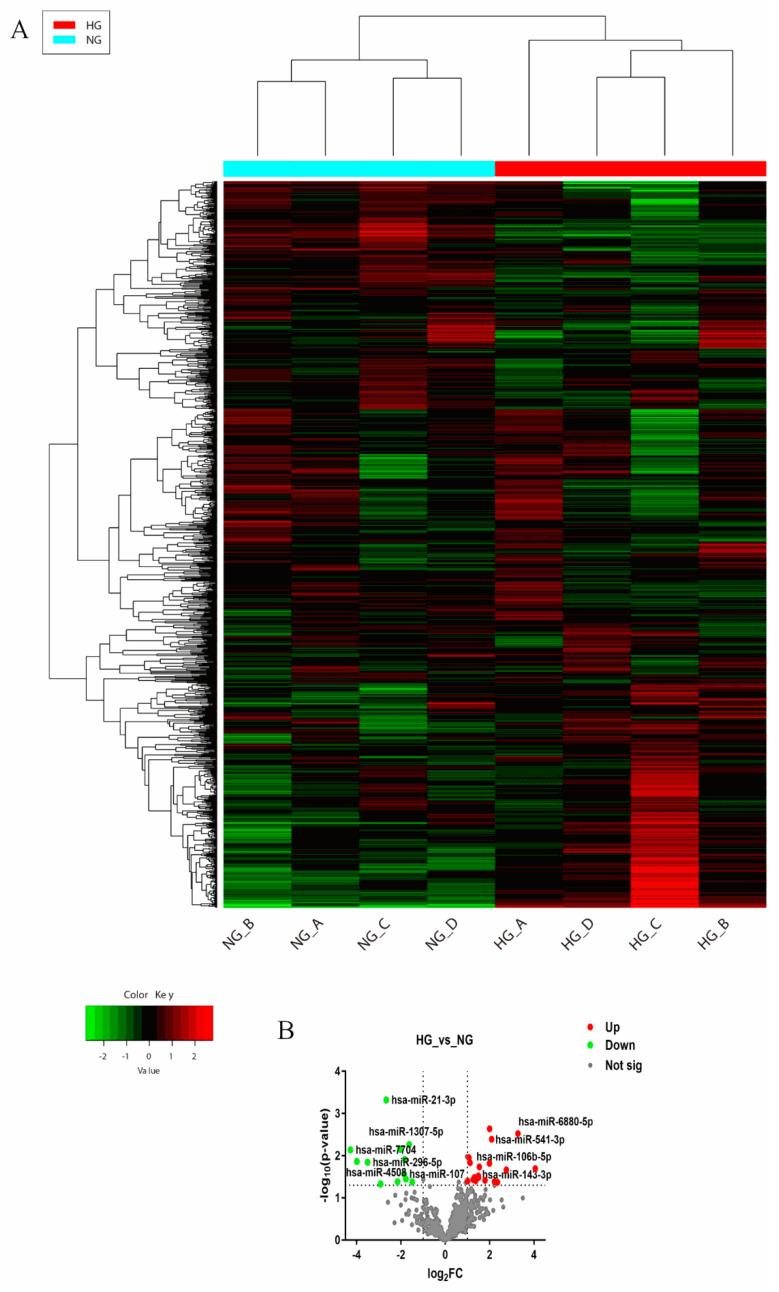
MiRNA sequencing. Eight samples were divided into four groups with one sample of NG and HG in each group. These samples were sequenced and subjected to bioinformatic analysis. (**A**) The heatmap indicates closeness between these groups. Red color reveals high expression of miRNAs, and green color reveals low expression of miRNAs. (**B**) The volcano plot showed the up- and down-regulated miRNAs in HG vs. NG. Red dots demonstrate the upregulated miRNAs, and green dots reveal downregulated miRNAs. The thresholds are defined as follows: upregulated miRNAs (Log_2_FC > 1, FC > 2, *p* < 0.05), downregulated miRNAs (Log_2_FC < −1, FC < 1/2, *p* < 0.05). HG—high glucose; NG—normal glucose; Not sig—not significant.

**Figure 4 genes-13-00640-f004:**
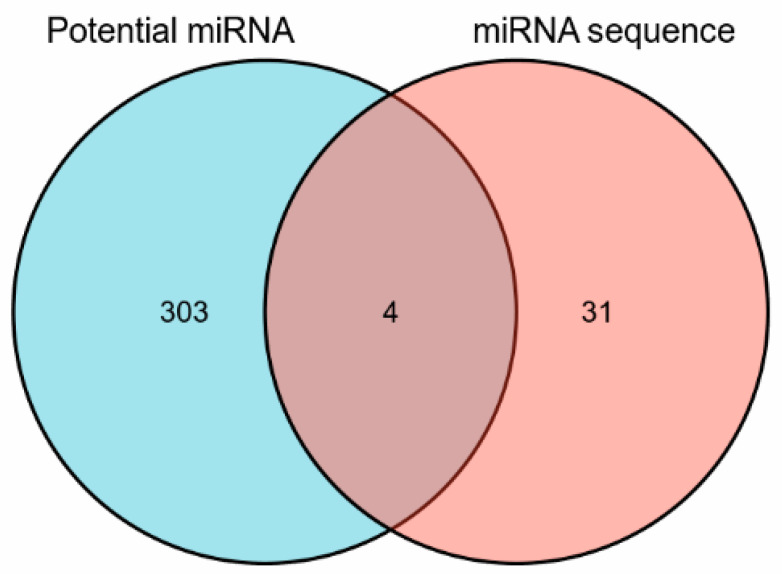
Overlapping miRNAs derived from text mining and from miRNA sequencing.

**Figure 5 genes-13-00640-f005:**
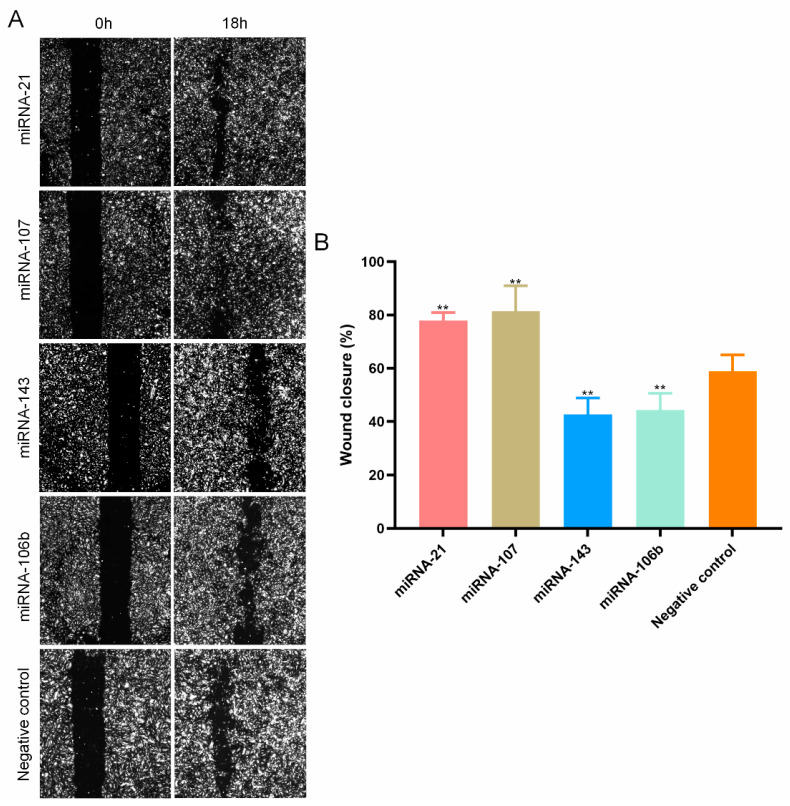
The overlapping four miRNAs influence the HUVECs migration. (**A**) microscopy images of HUVECs migration after transfected with the 4 miRNAs. (**B**) The wound closure (%) of HUVECs after transfected with the 4 miRNAs. ** *p* < 0.01. All experiments were performed independently three times.

**Figure 6 genes-13-00640-f006:**
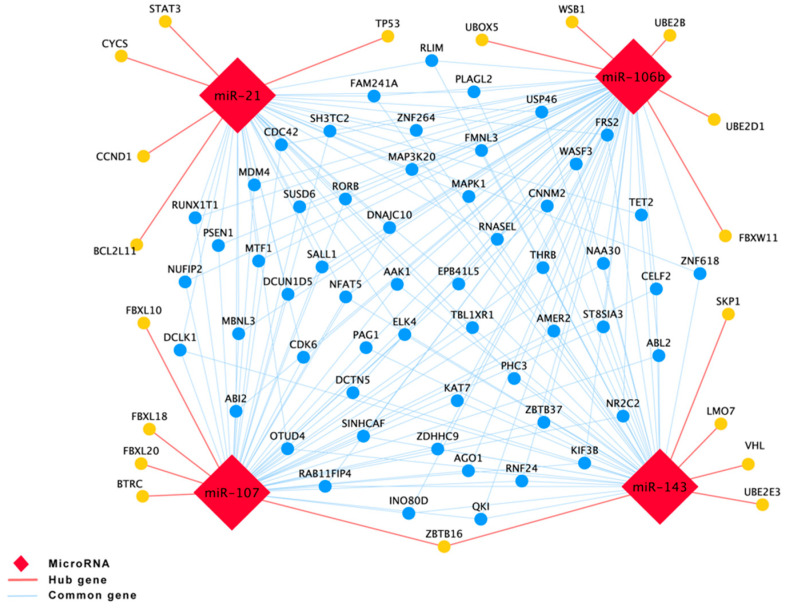
Bioinformatic analysis of targets and pathways of the 4 miRNAs. The protein–protein interactions of hub targets and the common targets among three and four miRNAs. (Cytoscape 3.7.2, https://cytoscape.org/index.html (accessed on 8 January 2022)).

**Figure 7 genes-13-00640-f007:**
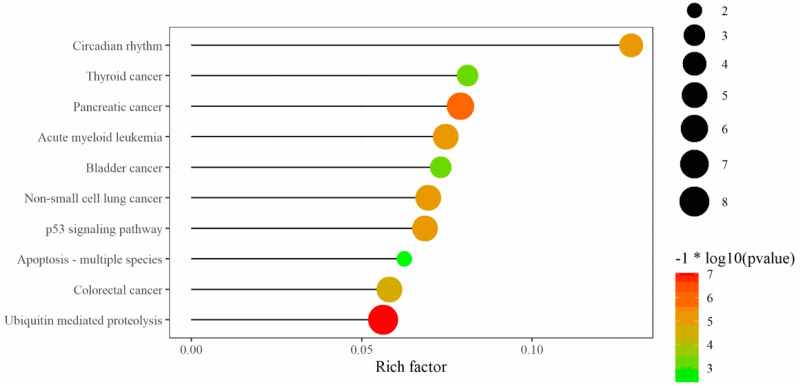
The KEGG analysis of the hub targets and the common targets shared between three and four miRNAs.

**Figure 8 genes-13-00640-f008:**
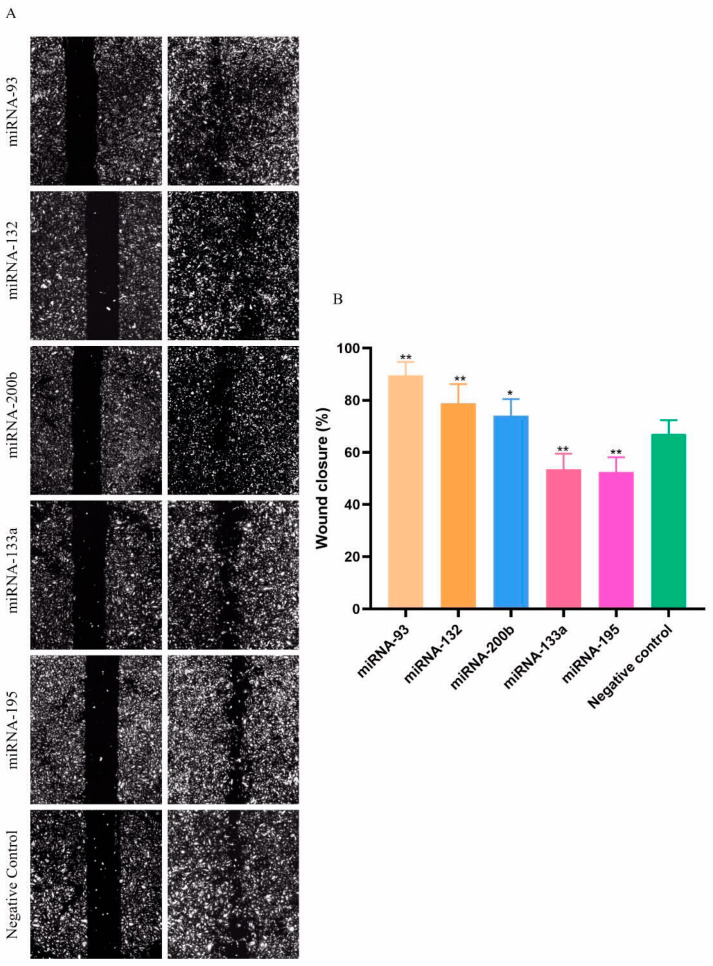
Unchanged expression of miRNAs may still influence HUVEC migration. (**A**) microscopy images of HUVECs migration after transfection with the 5 miRNAs. (**B**) The wound closure (%) of HUVECs after transfected with the 5 miRNAs. * *p* < 0.05, ** *p* < 0.01. All experiments were performed independently three times.

## Data Availability

The data presented in this study are available on request from the corresponding author.

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
