# Peer review of "MicroRNAs Influence the Migratory Ability of Human Umbilical Vein Endothelial Cells"

_genes, 2022, doi:10.3390/genes13040640_

Round 1

Reviewer 1 Report

In this article the authors used different approaches in order to identify miRNAs that are involved in migration of human umbilical vein endothelial cells (HUVECs). Combination of data derived from RNA sequencing and text mining led to a number of miRNAs which are involved in HUVEC migration and four of them identified to affect it. Several stipulations mentioned below, could be addressed to highlight the clarity and importance of the results presented:

  • In the method that is used for checking the migration, how can it be discriminated that the cells do not proliferate, so, for that reason the wound is closing and not just by the migration? There is need to elucidate that point and examine that the proliferation of the cells is not affected.
  • In fig 5A the wounds don’t have the same width in all cases, please correct.
  • In supplementary fig. 1, in fig. 5 and fig. 8 how many independent experiments have been done? Please refer it in the legend.
  • There has been done transfection in the cells, how were the levels of the miRNAs checked after the procedure?
  • Typos and grammar errors exist in the text. Below are some examples (not all) to help the authors in their corrections: Line 159 “obatined”, line 176 “namely”, line 190 “withmiR-21-3p”, line 194 “Using”, line 244 “still may still”, line 309 “This”, etc.

Author Response

Referee 1:

  1. In the method that is used for checking the migration, how can it be discriminated that the cells do not proliferate, so, for that reason the wound is closing and not just by the migration? There is need to elucidate that point and examine that the proliferation of the cells is not affected.

Response: We agree that it is important to also consider cell proliferation. However, migration assays were performed using 1% FBS containing medium. Hardly any cell doubling is observed under these starvation conditions. Furthermore, we have determined the doubling time of the Huvecs to be more than 48 hours under 10% FBS condition (Supplementary Fig 2). Since migration assays were followed only over the first 24 hours the vast majority of the gap closure can be attributed to cell migration. This information has now been included in the revised MS.

  1. In fig 5A the wounds don’t have the same width in all cases, please correct.

Response: We agree and have changed the corresponding images

  1. In supplementary fig. 1, in fig. 5 and fig. 8 how many independent experiments have been done? Please refer it in the legend.

Response: We agree. This information has now been included in the legends

  1. There has been done transfection in the cells, how were the levels of the miRNAs checked after the procedure?

Response: We agree that this a necessary control. We have now included information on the transfection efficiency of the labelled miRNA mimics (Supplementary Fig 3). This information has now been included in the revised MS.

  1. Typos and grammar errors exist in the text. Below are some examples (not all) to help the authors in their corrections: Line 159 “obatined”, line 176 “namely”, line 190 “with miR-21-3p”, line 194 “Using”, line 244 “still may still”, line 309 “This”, etc.

Response: We thank the referee and have omitted the typos.

Reviewer 2 Report

The manuscript submitted by Michael Keese, Vytaute Starkuviene and co-workers presents a set of analysis discussing the role of microRNA in regulation of migratory abilities of HUVECs.

The paper is very interesting. Introduction is brief and coherent, and gives a very good input into the molecular background of migration process, taking into account the current state of the art regarding miRNA. The methods applied are described with sufficient amount of detail. Presentation of data is clear and does not raise major comments.

I have major technical comment regarding the wound healing assay. In my opinion it is not legitimate to use the term migration as mitomycin C wasn’t use in the experiments to block the proliferation , and actually both processes are involved in covering the “gap”. I think the Authors should take it into account and reconsider the construction of the paper form the level of migration analysis, but also having in mind the effect of proliferation.

Author Response

Referee 2

I have major technical comment regarding the wound healing assay. In my opinion it is not legitimate to use the term migration as mitomycin C wasn’t use in the experiments to block the proliferation , and actually both processes are involved in covering the “gap”. I think the Authors should take it into account and reconsider the construction of the paper form the level of migration analysis, but also having in mind the effect of proliferation.

Response: We thank this referee for the positive overall assessment. We agree with his concern. This point has also been raised by referee 1. We have now provided data on the doubling time in the revised MS (Supplementary Fig 2). Since the doubling time is longer than the time span used for the observation of the gap closure, and the experiments are performed under 1% FBS medium conditions, most of the gap closure can indeed be attributed to the migration of the HUVECs

Round 2

Reviewer 1 Report

In this paper the authors tried to identify miRNAs that are involved in migration of human umbilical vein endothelial cells (HUVECs). The observations led to a number of miRNAs which are involved in the procedure and four of them identified to affect it.

Only minor comments can be made, for example, in Supp.Fig. 3, details can be added in the legend in order to explain further the method used.

Overall, the revised manuscript is improved as far as the way the results are presented, so it is easier for the reader to follow and the points that needed clarification have been covered in a high percentage. 

Author Response

Thanks for your valuable suggestion. We agree and have changed the legend of Supplementary Fig 3 as follows:

 Supplement Figure 3 Transfection efficiency of fluorescently labelled miRNA mimics. For the reverse solid phase transfection in the multi-well plates, the miRNA transfection solution was dispensed on 384 well plates. Fluorescently (Cy3) miRNA mimics were used (ThermoFisher, Germany, fluorescence tags on the 5`end). HUVECs were seeded in the pre-coated 384 well plates at a density of 800-1000 cells per well in 60 ul NG-ECGM and incubated for 24 hours. Hoechst 33342 was used to stain the nuclei. The fluorescently labelled miRNA mimics efficiently entered HUVECs cells. scale bar=50μm.

Reviewer 2 Report

I sustain my comment. I think the paper deserves to be published, however, I disagree with the authors. Yes, the double time of the population is longer that 24 h as presented on Supplementary figure 2, but the cells were not synchronized, are in the different phases of cell cycle, so within 24 h some of the cells will be able to divide. And it should be indicated in the work or an experiment with mitomycin C should be added.

Author Response

Dear Editor, dear referee,

we see your point and agree that we cannot exclude an influence of the proliferation for gap closure.

We have therefore added the following statement:

 "The current study has two limitations. Firstly cells were followed over 12 hours which is significantly shorter than the doubling time observed for HUVECs.  Since the cells were not synchronized for the migrations assays, to some extend cells may also undergo mitosis during the observation period. Therefore gap closure may not be attibuted to migration only, but also to some extend to proliferation. Secondly, since no experimental confirmation of the in-silico prediction of the hub genes has been obtained, ongoing studies will have to distinguish between migration and proliferation and...

We feel that we have hereby adequatly adressed the concern.

We hope that the revised MS is now suitable for publication.

Sincerely,

Michael Keese